**Perspective**

# Integrated fire management as an adaptation and mitigation strategy to altered fire regimes

I. Oliveras Menor [1,2] ✉, N. Prat-Guitart[3], G. L. Spadoni[1,4,5], A. Hsu[6], P. M. Fernandes [7],
R. Puig-Gironès [8,9], D. Ascoli [4], B. A. Bilbao[10,11,12], V. Bacciu[13,14], L. Brotons[15,16,17], R. Carmenta [18],
S. de-Miguel [15,19], L. G. Gonçalves[20], G. Humphrey[21], V. Ibarnegaray[22], M. W. Jones [6], M. S. Machado[2,23],
A. Millán[24], R. de Morais Falleiro[25], F. Mouillot[26], C. Pinto[22], P. Pons [8], A. Regos [15,27],
M. Senra de Oliveira[25], S. P. Harrison [28] & D. Armenteras Pascual [29]

Altered fire regimes are a global challenge, increasingly exacerbated by climate change, which modifies fire weather and prolongs fire seasons. These changing conditions heighten the vulnerability of ecosystems and human populations to the impacts of wildfires on the environment, society, and the economy. The rapid pace of these changes exposes significant gaps in knowledge, tools, technology, and governance structures needed to adopt informed, holistic approaches to fire management that address both current and future challenges. Integrated Fire Management is an approach that combines fire prevention, response, and recovery while integrating ecological, socio-economic, and cultural factors into management strategies. However, Integrated Fire Management remains highly context-dependent, encompassing a wide array of fire management practices with varying degrees of ecological and societal integration. This review explores Integrated Fire Management as both an adaptation and mitigation strategy for altered fire regimes. It provides an overview of the progress and challenges associated with implementing Integrated Fire Management across different regions worldwide. The review also proposes five core objectives and outlines a roadmap of incremental steps for advancing Integrated Fire Management as a strategy to adapt to ongoing and future changes in fire regimes, thereby maximizing its potential to benefit both people and nature.

## The challenge of altered fire regimes

We are experiencing fast changes in the timing, frequency, seasonality, size, intensity and severity of wildfires worldwide[1–3] when compared to historical ranges, i.e. altered fire regimes. Climate change modifies the fire weather[3] promoting extreme fire behaviour[4] and, in many regions, it has increased vegetation flammability[1,3], the frequency and intensity of wildfires[5] (Table 1). Overall, fire seasons lengthened by about 20% between 1979 and 2013[1]. Model projections suggest that burned area will increase by 9–14% by 2030 and 20–33% by 2050 even under the lowest emissions scenario[6]. Changing climate can increase the areas where fire occurs (i.e. fire-prone areas), impacting biodiversity, disrupting ecosystem functioning and endangering health, cultures and livelihoods, all of which amplify the vulnerability of ecosystems and human populations[7]. Extreme fire behaviour, characterised by fast and erratic spread, abnormally high intensity, and broad fire fronts, overwhelm civil protection and fire-fighting capacities[8]. These new conditions pose unprecedented challenges to the economy, society [9,10] and fire governance by reducing the time window for planned fire-use[11] and increasing firefighting costs[12] (Table 1).

Wildfires impact the climate system by emitting large quantities of greenhouse gases into the atmosphere: they currently account for 37.8% of the total emissions from natural sources and 16.9% of total natural and anthropogenic emissions[13]. Altered fire regimes are exacerbating wildfire-associated emissions. Forest fire carbon emissions have increased by 60% overall since 2001, driven largely by increased emissions from extratropical forests[14]. For example, wildfires in boreal forests released a record of 0.48 GtC in 2021, twice the average of the 2001-2018 period[15]. Fires in Canada emitted 1.3 Pg $CO_2$ (0.39GtC) in 2023[16], double the total $CO_2$ equivalent emissions for this country in 2021 (estimated to be 0.67 Pg $CO_2$ (0.2 GtC)[17]).

**Table 1 | How IFM, through the proposed core objectives and actions, can be used as a mitigation and adaptation tool to altered fire regimes, and what are the current challenges and priorities for future directions**

| Aspects of Altered Fire Regime | IFM core objectives | IFM Actions | Mitigation and adaptation outcomes | Current challenges | Priorities and future directions | References |
|---|---|---|---|---|---|---|
| **Changes in fire weather** | | | | | | |
| - Shifted fire weather<br>- Increased frequency and intensity of wildfires<br>- Extreme wildfire events | - Mitigate wildfire risk<br>- Carbon emission abatement | - Planned and prescribed fuel treatments for wildfire risk reduction and to minimise fire severity<br>- Adaptive management for building landscape mosaics<br>- Reflect climate change projections in fire risk, improve accuracy of predictive models, understand ecology of key species and key services of local ecosystems<br>- Maintain or increase carbon stocks in vegetation and soil through carbon abatement projects that incorporate traditional ecological knowledge | - Landscapes resilient to changing fire dynamics (as a climate adaptation strategy).<br>- Reduction in vulnerability and impacts of wildfires on society, economy, and ecosystems (for climate resilience)<br>- Enhanced well-being and ecosystem stability (as a climate change adaptation outcome)<br>- Decreased greenhouse gas emissions through effective wildfire management and landscape conservation | - Availability of technologies working at fine spatial and long temporal scales relevant for monitoring changes in fire regimes<br>- Models that accurately forecast changes in fire risk and fire regimes | - Dynamic, data-driven models for fire risk assessment, incorporating real-time environmental data and predictive analytics<br>- Local capacity building | 2,3,5,120 |
| **Changes in fire behaviour** | | | | | | |
| - Increased atmospheric instability<br>- Increased fuel availability<br>- Increasingly common extreme wildfire behaviour | - Mitigate wildfire risk<br>- Carbon emissions abatement<br>- Ecological conservation/restoration<br>- Enhance landscape resilience | Adaptive management identifying new priority areas for fire protection<br>- Planned fuel treatments for wildfire risk reduction and to minimise fire severity | - Minimise risk of social, ecological and economic damage from extreme wildfire events<br>- Minimise greenhouse gas emissions | - Accurate prediction of wildfire occurrence and their behaviour<br>- Identify unknown fire effects to society and to ecosystems associated to shifted wildfire behaviour | - Identify new fire-prone areas, and areas likely to burn with extreme fire behaviour<br>- Improved wildfire spread models that incorporate prolific spotting and fire-atmosphere processes | 4,7,119 |
| **Decrease of landscape resilience** | | | | | | |
| - New fire-prone areas<br>- Water quality degraded by wildfires<br>- Water security<br>- Increased costs of fire suppression | - Enhance landscape Resilience<br>- Mitigate wildfire risk<br>- Carbon emission abatement | - Adaptive management, frequently update fire risk assessments to identify new areas susceptible to fires and adapt to new conditions<br>- Incorporate traditional fire uses and needs, as well as traditional ecological knowledge<br>- Develop adequate landscape planning strategies<br>- Manage vegetation to reduce wildfire risk | - Achieved diversity of land uses that build resilience against altered fire regimes by:<br>- *Enhanced human wellbeing*<br>- *Optimised ecosystem health*<br>- *Secured and safeguarded water resources*<br>- *Reduced risk and impact of uncontrolled wildfires* | - Moving from 'reactive' to 'adaptive' management<br>- Lack of 'one size fits all' solutions, context dependency of policies and management plans<br>- Weak legal frameworks to develop adaptive management practices<br>- Loss of traditional ecological knowledge<br>- Post-fire impacts data availability | - Multi-actor cooperation and collaboration<br>- Legal frameworks that enable adaptive management policies<br>- Economic resources to build resilient landscapes based on prevention, adaptation and mitigation actions<br>- Research: coupling fire likelihood with water flow and erosion models<br>- Local capacity building and engagement<br>- Adaptive vulnerability assessments that translate into effective adaptation and mitigation actions for local populations | 33,67,87,88,121 |
| **Biodiversity loss** | | | | | | |
| - Decreased vegetation resilience<br>- Loss of biodiversity and ecosystem function<br>- Alien plant invasion | - Ecological conservation and restoration<br>- Promotion of local livelihoods and knowledge | - Incorporate traditional ecological knowledge into adaptive management<br>- Establish clear and achievable ecological conservation objectives<br>- Foster natural ecological regeneration pathways<br>- Prevent the encroachment of invasive species<br>- Promote post-fire species refugia | - Enhanced biodiversity conservation and restoration, securing ecosystem services like pollination, water provision and quality<br>- Reduction in the emergence of new fire-prone ecosystems | - Know desired level of fire to maximise biodiversity and ecosystem function in fire-prone ecosystems<br>- Measure responses to fire in trophic networks and across trophic levels<br>- Determine species vulnerability to fire in new fire-prone ecosystems<br>- Decipher fire-invasive impacts positive feedbacks<br>- Forecast magnitude of fire-driven ecosystem change under different climate scenarios | - Post-fire natural restoration planning<br>- Fire ecology of new fire-prone areas<br>- Research: more field data; more investments on long-term research programs | 98,121–124 |

**Table 1 (continued) | How IFM, through the proposed core objectives and actions, can be used as a mitigation and adaptation tool to altered fire regimes, and what are the current challenges and priorities for future directions**

| Aspects of Altered Fire Regime | IFM core objectives | IFM Actions | Mitigation and adaptation outcomes | Current challenges | Priorities and future directions | References |
|---|---|---|---|---|---|---|
| **Changes in fire weather** | | | | | | |
| **Pressures on society and governance** | | | | | | |
| - Poor air quality due to extreme wildfires<br>- Reduced windows for planned fire-use<br>- Poverty, increase in inequalities and displacement | - Mitigate wildfire risk<br>- Enhance landscape resilience<br>- Promotion of local livelihoods and knowledge | - Adaptive management plans<br>- Effective fuel management to reduce wildfire risk<br>- Community engagement in management and planning<br>- Local capacity building<br>- Establish community-based monitoring systems for early wildfire detection and rapid response<br>- Multi-stakeholder cooperation and knowledge sharing<br>- Ensure knowledge traceability and recognition | - Fire-adapted communities<br>- Reduced vulnerability and enhanced resilience to wildfires<br>- Establishment of robust/strong adaptive governance systems to manage resilient landscapes | - Quantification of health impacts across populations associated to indirect Effects of wildfires and/or extreme weather<br>- Adequate governance for mitigating impacts of extreme wildfires in vulnerable population<br>- Availability of resources to develop actions<br>- Disappearance of local and indigenous knowledges | - Health impact studies<br>- Strong legal and governance frameworks<br>- Ensure flexible, responsive policies that accommodate changing conditions and knowledge<br>- Effective stakeholder collaboration and knowledge sharing<br>- Provide the right channels and means for knowledge exchanges<br>- Community-based capacity building and empowerment<br>- Integration of IFM into broader climate change policies | 25,67,81,83 |

The extreme fires in Australia during the summer of 2019 emitted 0.71 Pg $CO_2$ (0.23 GtC)[18].

Altered fire regimes result from direct (e.g. ignition sources, land-use change, fire management policies[19]) and indirect anthropogenic actions (e.g. climate change[20,21]). Over the last century, human activities have profoundly modified ignition patterns and landscape flammability through land use change, fire suppression policies[22] based on excluding all types of fires (even in fire-prone regions), and the disappearance of fire uses and practices linked to traditional ecological knowledge[23]. These changes, combined with extreme wildfire activity in recent years, have revealed the limitations of prevalent fire policies focused on emergency response and fire-suppression, underscoring the need for more integrated, effective, and holistic fire management strategies.

Integrated fire management (IFM) – an holistic approach that integrates management, ecology and society – may help address the consequences of past fire suppression policies and challenges posed by altered fire regimes by applying a nuanced understanding of fire's ecological and cultural dimensions. However, its applicability though has not yet been systematically assessed.

This review examines current fire management practices, with a focus on IFM as an adaptation and mitigation strategy to altered fire regimes. We review the concept of IFM, assess the progress and challenges in its implementation across different regions worldwide. We then propose five core objectives and a roadmap of incremental steps for implementing IFM as a strategy to adapt to ongoing and future changes in fire regimes, and maximise the potential of IFM to provide benefits to people and nature.

## Shifting paradigms: from fire suppression to integrated fire management

Over the past few centuries, most fire management strategies have predominantly focused on emergency responses and fire suppression, and on fire bans as the only way of wildfire prevention. Active, organised fire-fighting emerged in the 19th and 20th centuries in Europe, Australia, and the North America, and spread to other parts of the world aided by colonial influence[24]. This fire suppression approach (reinforced since the 1970's with aircraft water firefighting) did not recognize the ecological role of fire or its local cultural and social significance[24,25], and failed to understand that numerous local forms of fire use were acts of fire prevention or reduction of fuel build-up[26]. Socio-economic and land-use changes in the 20th century—such as rural abandonment (i.e., migration from rural to urban areas) in Europe, the growth of industrial forestry and agribusiness in South America, and the expansion of residential areas that increase the wildland-urban interface in North America and Europe — have strengthened the perception of fire as a universal threat to society and ecosystems. This has led to a widespread perception that fire should be suppressed at all costs, regardless of the type of fire. This misleading belief that fire is a controllable artefact, rather than a natural process intrinsic to ecosystem dynamics, has been enhanced by media misinformation[27] and biased debates[28,29].

Legislation criminalising traditional and Indigenous fire practices[29,30], coupled with large investments in suppression technologies, have bolstered anti-fire perceptions worldwide, even where fire removal has had negative impacts on ecosystems or local livelihoods that are embedded in traditional fire management practices, exacerbating social disadvantages and inequality. Fire suppression policies have been predominantly aimed at reducing burned areas without addressing fire prevention or post-fire recovery, leaving a legacy of homogeneous landscapes with high fuel loads that enhance the risk of large and catastrophic wildfires[26,31,32].

The fire management community's acknowledgment of the long-term ineffectiveness of suppression-centric policies in preventing destructive fires has led to the development of prevention-oriented programs[33,34]. Some of these include active ecosystem management (e.g. prescribed burning programs) to minimise wildfire risk[35]. Other approaches (e.g. holistic fire management'[36], 'Indigenous fire stewardship'[37], 'intercultural fire management'[25], 'cohesive management'[38] and 'community-based fire

management'[39,40]) aim at revitalising local and Indigenous knowledge as a mechanism for fire management, territorial care, and place making.

The concept of IFM, initially coined in the 1970s[41,42], marked a shift towards a more comprehensive strategy that emphasised fire prevention and preparedness, and promoted ecological recovery after fire, moving beyond emergency responses to fire. Myers et al.[43] defined IFM as an approach that integrates three fundamental dimensions of fire: management (encompassing prevention, suppression and fire use), ecology (focusing on key ecological attributes of fire) and culture (considering both the socio-economic and cultural imperatives for fire use along with the negative impacts that fire can have on society). The concept of IFM defined by Myers et al.[43] has gradually gained recognition among fire practitioners and scientists because of its potential to deliver effective, efficient and equitable fire management. IFM builds on synergies between multiple goals such as wildfire risk mitigation, biodiversity conservation and restoration, landscape resilience, improving livelihoods and preserving knowledge and cultural values[44].

IFM is context-dependent and therefore must be planned in accordance with the local socio-ecological specificities and management objectives (Fig. 1). The achievement of IFM objectives requires an understanding of the ecology of the system (including fire-sensitive ecosystems), the services the ecosystems provide, and how these properties are affected by fire types, from the perspective of multiple actors and sectors. However, despite the attention it has gained and the many IFM initiatives that have flourished around the world, the lack of a formal definition, standardization and regulation – which is the result to a lack of political will - has kept IFM initiatives at relatively small scales. This is a missed opportunity, especially in face of the climate changes that are changing the flammability of many landscapes and, together with other anthropogenic drivers, profoundly altering fire regimes worldwide. IFM can not only help people to re(learn) to live with fire, and with wildfires, it can also revitalise traditional and Indigenous knowledge, maintain ecosystems services and assist in adaptation and mitigation policies to the emergent new fire regimes.

## Advances in developing and implementing IFM

Countries are at varying stages in the adoption and implementation of IFM: some countries have made rapid advances by providing adequate frameworks and establishing IFM programs, others are still in the early stages of adoption (Fig. 1, Supplementary Table S1). Despite challenges in implementing and achieving conservation and management goals, most of these initiatives represent a step-change towards reducing wildfire risk, and promoting ecological and social integration, and cultural acknowledgement (Fig. 1, Supplementary Table S1). As an example, Australia has pioneered carbon mitigation efforts based on Aboriginal fire-use practices, predominantly administered and managed by Aboriginal communities living in the region[45]. The pioneer Western Arnhem Land Fire Abatement (WALFA) project in northern Australia, a partnership involving Aboriginal peoples and the industry sector, focuses on savanna-burning for the carbon market[46]. During its initial years, WALFA decreased mean annual emissions by 38% relative to the baseline[47]. In California, prescribed burning on Yurok and Karuk tribal lands is part of their forest carbon-offsets selling scheme[48].

Fire management in several African nations is still largely influenced by colonial legacies, characterised by top-down centralised governance, suppression-centred policies, and under-resourced approaches[49]. Consequently, despite the prevalent and culturally significant practice of community-based fire use as a land management tool across many African communities (e.g. refs. 49,50), there is often a lack of adequate institutional support for effective use of resources for IFM that would bring both ecological and societal benefits including contributing to climate change adaptation. However, where resources are available, IFM can achieve several goals and balance needs. For example, in Kafue National Park (Zambia), IFM supports a sustainable fire regime that fosters ecotourism with open-space maintenance, providing job opportunities, and allowing farmers to use fire for pasture maintenance, pest control, crop residue disposal, and maintain high-yield crops (Fig. 1, Supplementary Table S1)[51].

In Ethiopia, efforts are underway to develop a national IFM strategy that acknowledges traditional uses of fire and develops a national system aimed at reducing wildfire risk, including an early warning system. An important step forward was made in 2023 with the acknowledgement that fire suppression enforcement was inadequate and ineffective, and the identification of priorities, key actors and steps needed to develop an effective national IFM strategy[52,53].

There are several initiatives implementing IFM in Latin America. In Brazil, IFM was progressively introduced in Indigenous Territories (ITs) and Conservation Units since 2013 through initiatives including the establishment and training of Indigenous fire brigades, the integration of traditional Indigenous fire-use practices into official fire management plans, the development of a governance structure established through signed agreements between Indigenous Communities and federal agencies, and the advancement of research and monitoring through scientific programs[54] (Fig. 1, Supplementary Table S1). In July 2024, Brazil was the first Latin-American country to approve a Federal Law on Integrated Fire Management[55]. In the Brazilian regions where IFM is used, it has successfully reduced the annual burned area affected by wildfires by almost 20%[54]. In Mexico, IFM principles and actions have been progressively implemented in several places since the early 2000s[56] and the country incorporated IFM in the new national Law on Sustainable Forest Development (approved in 2018 and enacted in 2021[57]). Currently Mexico has a nation-wide fire management plan that recognizes the ecological and social role of fires, and establishes actors and responsibilities for different aspects of the plan[58]. In the Amazonian basin, OTCA (Organización del Tratado de Cooperación Amazónica) is an example of a supra-national governance organisation that in 2021 promoted a Memorandum of Understanding on Integrated Fire Management[59] that has been signed by the OTCA member countries (Bolivia, Brazil, Colombia, Ecuador, Guyana, Perú, Surinam, Venezuela). This agreement has been a precedent for several national initiatives promoting IFM. In Bolivia, indigenous and traditional communities in the Chiquitania region have been part of a community-based fire initiative that has promoted and implemented IFM practices since 2011[40]. Similarly, Venezuela's INPARQUES Fire Brigade has embraced IFM with intercultural practices since 2015, building on the integration of Indigenous and scientific knowledge with the technical expertise of its forest firefighting teams[60].

In Southeast Asia, more integrated approaches to the classical fire management of peatland fires have been under consideration. When wildfires are powered by the combustion of deep layers of drained peat, the classical fire suppression strategies are insufficient to control and extinguish fires. In the Indonesian regions of Riau, Sumatra (Jambi and Palembang) and Central Kalimantan, multiple bottom-up initiatives have emerged in reaction to this situation by moving from the existing fire suppression model towards community-based fire prevention, preparedness, and suppression strategies[61,62]. These strategies have a positive impact in reversing land abandonment caused by the sale of land to external investors with exploitation interests[63] and support a transition towards sustainable livelihoods, where fire is used as a cost-effective method for localised clearing of land for agriculture and fishing. However, despite the interests of local communities in IFM, their engagement remains limited due to a lack of institutional and financial support[61].

In Europe, fire management programs using prescribed burning have been implemented since the 1980s by professional organisations and networks. Examples of fire management programs evolving into more holistic IFM practices are found in Portugal, France, Spain, Italy, and Sweden[64,65]. For example, a prescribed burning program was initiated in the Pyrénées-Orientales region (southern France) in the 1980s to manage shrub encroachment and restore natural grasslands for pastoral purposes[66]. This program has evolved towards serving multiple objectives: open spaces for pastoralist uses, biodiversity conservation of key flora and fauna species, management of game fauna, fire-managers training, and wildfire risk reduction (Fig. 1, Supplementary Table S1). The governance structure

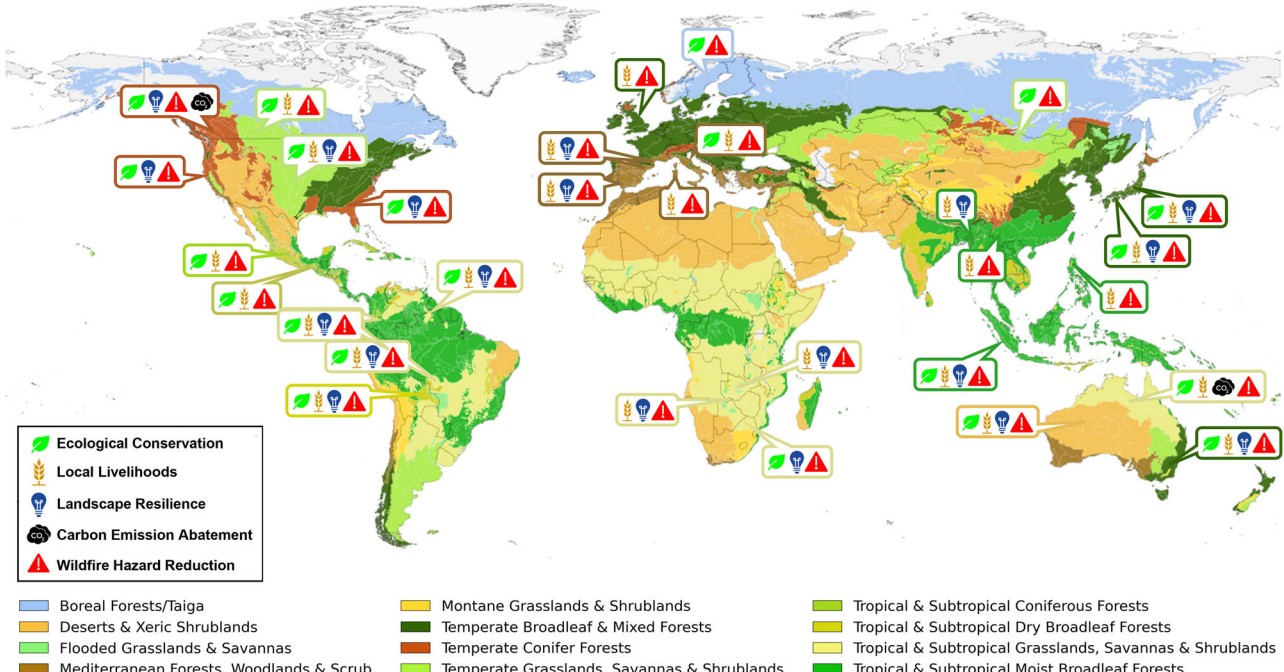

**Fig. 1 | Locations where IFM are being implemented following some of the IFM goals for adaptation and mitigation to altered fire regimes.** Some geographical locations may contain several initiatives. See Supplementary Table S1 for a complete list.

includes national and regional agencies as well as the local communities who are partly responsible for prioritising areas to burn and for executing the prescribed burns[66].

## The grass is not always greener: current constraints on IFM

Fire management is a 'wicked' problem[67] encompassing many social, ecological, economic and governance layers, for which there is no 'one size fits all' solution. This complexity has slowed the shift from emergency-focused fire management policies. Strong fire risk aversion perceptions and a lack of regulatory frameworks have hindered the development and implementation of IFM at large scales. IFM is inherently a multi-faceted approach, but the terminology 'Integrated fire management' has often been applied to management strategies limited to prescribed burns for wildfire risk reduction, without fully accounting the social or ecological perspectives.

The perception that IFM initiatives necessarily involve the use of controlled or planned fire have also given rise to criticism about the impacts of these fires on air and water quality[68]. Whenever fire is used, a careful consideration of the risks to human health and air quality is needed. Furthermore, the ecological impacts of planned burns must also be considered, especially when fire is not a natural component of the ecosystem, or when planned burns do not have ecological objectives and are carried out at a different time from the natural burning season in that ecosystem. For example, planned burns for wildfire risk reduction or carbon abatement potential can severely impact the phenology and life history of communities[69]. The introduction of early season burning in African savannas aimed at reducing wildfire risk and greenhouse emissions[70], highlights the need for caution. African savannas include many different vegetation types from open ecosystems with perennial grasses to ecosystems dominated by shrubs and deciduous woody vegetation. Vegetation dominated by woody plants is often too moist to burn in the early dry season, causing lower combustion rates and higher emissions. Many African ecosystems have experienced significant woody encroachment over the last few decades[71,72], and early-season burns will thus be less effective in reducing fuel loads.

Progress in establishing regulatory frameworks that fully embrace IFM and its socio-ecological importance has been slow in most places of the

world. For example, in Bolivia, Colombia, and Venezuela bills to implement new national IFM policies have been proposed but await formal approval. There are no policies or standardised IFM programs in any European country, and fire management activities in many countries (e.g. Portugal, Spain, France) tend to aim exclusively at reducing wildfire risk[73]. This is also the case for the many countries in the African and Asian continents. In Europe, an added challenge is the lack of detailed knowledge about traditional/cultural burning (but see refs. 74–77), how it was practiced and to what extent it is still being used. Fire-exclusion policies and land abandonment have led to increased fuel accumulation across Europe, creating fundamental social and economic changes that are a challenge to developing IFM programmes[78].

## Setting IFM as an adaptation and mitigation strategy to altered fire regimes

IFM – by holistically integrating environmental, sociocultural, economic and management perspectives – could be a powerful strategy to mitigate and adapt to changing fire regimes. However, to date, IFM programmes are still context specific. The recently updated IFM Voluntary Guidelines from the Food and Agriculture Organization of the United Nations (FAO)[79] provide a good basis of what needs to be considered when developing an IFM program, but these guidelines are rather broad. Here, we advance by proposing five core interconnected objectives that IFM programmes should adopt to adapt their territories, peoples and nature successfully to new fire regimes and mitigate the impacts of destructive wildfires:

### Enhance landscape resilience

Resilient landscapes can maintain, renew and strengthen their fundamental qualities despite disturbances or ongoing changes[80]. Climate change impacts fundamental qualities such as water and food security, or wildfire risk. IFM can mitigate these through the integration of climate projections into fire risk prediction, the development and implementation of holistic land management practices that include fuel treatments, and through community engagement and coordinated collaboration among fire prevention and emergency response agencies[81,82]. This can assist adaptation through bottom-up approaches that incorporate traditional knowledge, education, and understanding of the local socio-ecological and economic contexts to

design and implement integrated landscape planning strategies through preparedness, response and recovery actions. Such an approach should make use of a combination of early warning systems, community engagement, land use and urban planning, sustainable schemes for fuel management, emergency response planning, research, and collaboration among various stakeholders[81,83]. Lastly, any IFM actions need adequate metrics to measure effectiveness in a given landscape and facilitate adaptations to changing conditions.

### Promotion of local livelihoods and knowledge

Combining knowledge and understanding and applying traditional fire-use practices not only enhances ecosystem resilience, mitigates fire severity, and protects biodiversity but also aligns modern conservation efforts with centuries of ecological and cultural practice. Local knowledge holders, who witness current climate change impacts alongside historical fire and landscape uses, are deeply tied to the environment[84,85]. It is estimated that Indigenous Peoples steward or hold tenure rights over at least 37% of the Earth's natural lands, containing more than one-third of the world's intact forests[86].

Traditional knowledge can promote 'cool' low-intensity fires that prevent 'hot' uncontrolled wildfires and facilitate adaptation to new fire regimes in ecosystems that do not cope with high intensity fires. It can also enhance ecosystem resilience via agro-silvo-pastoral activities, and facilitate post-fire restoration by selecting appropriate species and sites, and invasive species control. Integrating local knowledge with modern scientific and management methodologies improves ecological outcomes, reduces economic costs, and fosters social acceptance[87–89]. In some systems, the use of low-intensity fire can enrich the soil with nutrients, while rotational burning practices can ensure that ecosystem functions have sufficient time to recover, both crucial for sustaining small-scale subsistence agriculture and food security[90,91]. Planned fires can mitigate wildfire severity, safeguarding crucial resources such as forests and grazing lands essential for local communities' needs, including livestock forage, medicinal plants, and timber. Fire management can also create economic opportunities by increasing food production for local trade, attracting ecotourism through biodiversity conservation activities, and enabling sustainable harvesting of non-timber forest products.

The role that local communities play in preserving biodiversity and regulating fire within safe boundaries should be rewarded, for instance, by accounting for avoided damages to other ecosystem services and acknowledging how this contributes to climate change adaptation[92].

### Ecological conservation and restoration

In response to the increasing biodiversity crisis[93], the United Nations has declared 2020-2030 as the Decade of Ecosystem Restoration. This requires consideration of options for restoring biodiversity and ecosystem function. IFM initiatives can be part of these options because they support climate change adaptation by moderating extreme fires and buffering their impacts on ecosystems and biodiversity[94].

Ecosystems are inherently complex systems and IFM has to account for site-specific conditions to achieve ecological goals, including identification of key species and the adaptability of different species to fire. Planned burning (e.g. patch mosaic burning), for example, creates mosaics of different habitats and promote species diversity and natural ecosystem regeneration after fires[95,96]. IFM can also promote natural regeneration by preserving potential refugia and conserving mature ecosystems that are essential to maximise biodiversity and resilience[97,98]. Preserving riparian corridors, for example, provides natural pathways for species dispersal[99] and may provide refuges during droughts or wildfires[100].

IFM aimed at reducing fuel loads and continuity can help maintain biodiversity and ecosystem functioning in open habitats. In Southern Europe, for example, land abandonment has negatively affected open habitat species, such as wet grassland or early successional species. IFM can help restore these ecosystems while promoting greater fire resistance and climate-adaptive landscapes[77].

### Mitigation of wildfire risk

IFM can mitigate wildfire risk through fuel management strategies targeted at changing vegetation composition and structure, reducing fuel load, or reducing vegetation continuity[44]. The choice of strategy depends on the local context and management goals. Fuel treatments to reduce the accumulation of surface fuel (litter, grasses, shrubs) include planned fire-use (from fire-use by communities to prescribed burning), the mechanical or manual clearing of understorey vegetation, and grazing. Silvicultural practices — pruning and thinning, variable retention harvesting, closer-to-nature forestry — may be designed to either decrease the likelihood and intensity of a crown fire or decrease understorey flammability through increased shading and sheltering.

Linear fuel breaks are designed to limit the spread of wildfires, but their effectiveness is highly variable[101]. Traditional pastoral and aboriginal burning in Europe[102] and Australia[103] respectively, has been shown to decrease wildfire size substantially. Similarly, area-wide fuel treatments (either prescribed burning, thinning, or both) have been shown to mitigate wildfire extent and severity[104–106]. Although extreme fire weather may override the effects of fuel treatments on wildfire spread, environmental and societal benefits will persist as long as heat release, which correlates with fuel load, remains lower than in untreated areas[107]. Lastly, fuel treatments may have some harmful ecological consequences (e.g. landscape fragmentation) that could be avoided or mitigated if fuel treatments were designed and maintained in a way that are not detrimental to other objectives.

### Carbon emission abatement

Increased or stabilized carbon storage may arise as a by-product of forest fuel-reduction treatments, depending on the amount and frequency of carbon removal and the effects on future wildfire probability and severity[108]. Mitigation of carbon emissions can be an explicit or even primary fire management goal if it also brings socio-ecological benefits. In many fire-prone areas, changing fire regimes – such as later-season burning – have led to higher fire intensity[70]. Such areas can be managed, depending on the local ecological context, by early to mid-season burning that is less intense, emits less greenhouse gases and decreases the risk of extreme and large wildfires in the late dry season[47,70].

Thus, carbon emission abatement resulting from IFM can open a window of opportunity for compliance and voluntary carbon market schemes that can benefit local communities by enhancing social resilience, however this is context dependent. While this is a potential added opportunity, it is not a panacea: application in Australia remains limited to a few vegetation types[47], and is hindered in Eastern and Southern Africa by multiple challenges associated to asymmetries of the IFM goals (e.g. biodiversity restoration versus carbon emission abatement[39,109,110]) as well as between local governance systems and formal institutions[111]. Carbon abatement programmes should be assessed according to a country position statement, which prioritises climate change adaptation and mitigation alongside biodiversity conservation and the local communities involved in such initiatives.

### Roadmap for countries

Building on the insights synthesized in this review, we propose a roadmap to implement Integrated Fire Management (IFM) at national or regional scales. This framework integrates the five core objectives described above, addresses diverse fire scenarios, engages multiple stakeholders, and considers potential risks to ensure informed and adaptive decision-making (Fig. 2).

1. *Assessment and planning phase*:
- Conduct a comprehensive assessment of current and future wildfire risk and vulnerability to extreme events, considering both ecological and socio-economic factors[112].
- Establish the desired level of fire to maximise biodiversity and ecosystem function, while allowing the use of fire for pastoral, agricultural and cultural purposes.

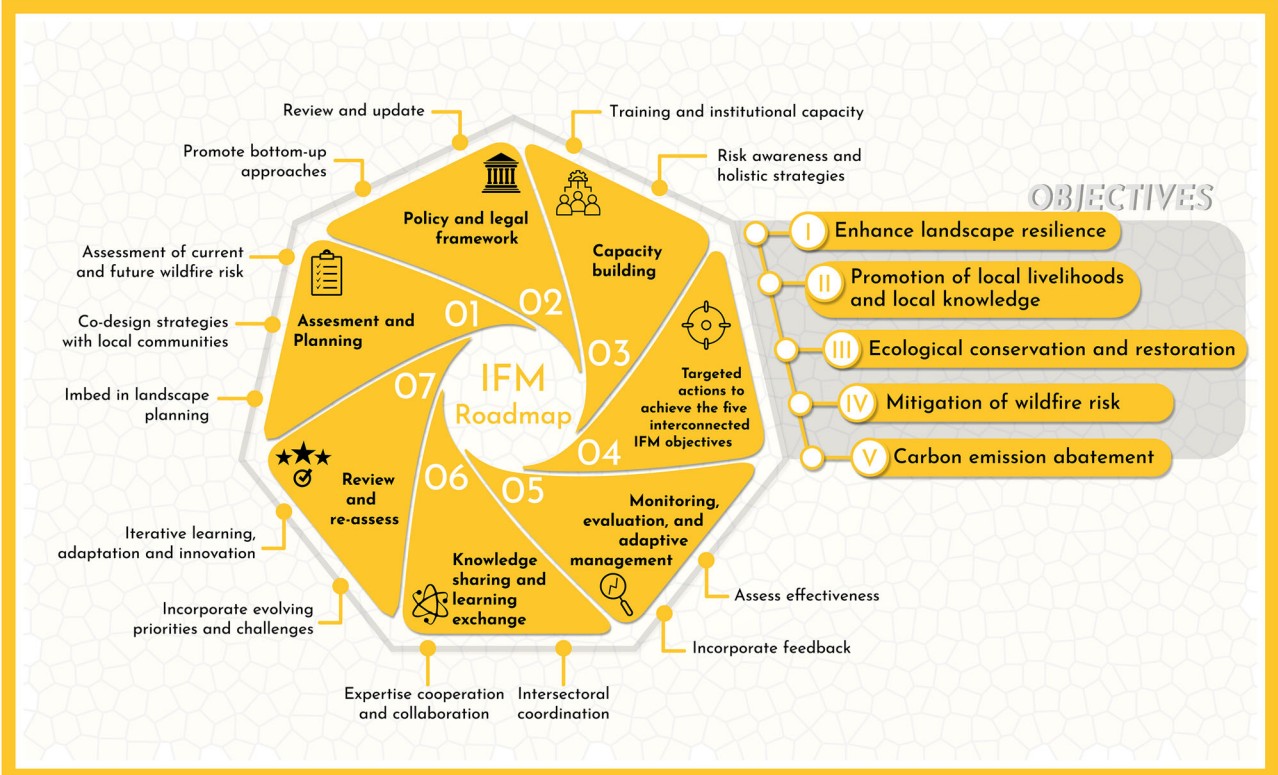

**Fig. 2 | Roadmap to developing IFM strategies aimed at adaptation and mitigation to altered fire regimes.** This roadmaps involves seven core steps (see section 'Roadmap for Countries') and five main objectives (see section 'Setting IFM as an adaptation and mitigation strategy to altered fire regimes' for a description).

- Co-design IFM strategies with clear short-, mid- and long-term objectives, targets, and actions, and identifying potential pitfalls and undesired side effects (e.g. increased $CH_4$ emissions from grazing, ecological damage linked to planned burning outside the fire season or where fire is not a natural agent).
- Embed IFM strategies as part of wider landscape and territorial planning[81], enhancing rural communities' cultures, values and livelihoods that contribute to the preservation of fire knowledge, wildfire risk reduction and biodiversity conservation.

2. *Develop a policy and legal framework*
- Review and update existing policies and legal frameworks to integrate IFM principles and practices ensuring that bottom-up practices (e.g. traditional fire use) are covered and supported by the legal framework.
- Establish regulations and incentives to promote IFM adoption and incentivize sustainable fire management practices, minimising risk-aversion to fuel treatments.
- Ensure that emergency response policies encompass IFM. Decision-making during large incidents must take into account different levels of governance, knowledge and community-based fire management.

3. *Capacity building*:
- Invest in capacity building for IFM implementation, including training programs for fire practitioners, land managers, policymakers and members of the local community.
- Train emergency responders, in particular decision-makers, in fire analysis/assessment, fire ecology and fire use, so that they have a better understanding of the impacts their decisions on local socio-ecological dynamics.
- Raise awareness of emerging challenges and propose holistic adaptation and mitigation strategies that combine local knowledge with scientific and technical understanding[113].

4. *Address risks of undesired effects and implement actions to achieve the five interconnected IFM objectives* (Table 1):
- Implement landscape-scale approaches to enhance landscape resilience to wildfire and climate change impacts. This may include ecosystem-based adaptation measures such as natural buffer zones and green infrastructure.
- Promote local livelihoods and local knowledge by supporting the revitalization of Indigenous and traditional fire management practices. This should involve incorporating traditional ecological knowledge into IFM strategies and developing sustainable livelihood opportunities, such as eco-tourism[114], agro-silvo-pastoralism[83], and sustainable forestry[115].
- Prioritise the conservation of key species within the landscape acknowledging their socio-ecological role. This involves identifying and prioritising biodiversity hotspots, biota refuges and critical ecosystems and implementing sensible habitat restoration programs and promoting ecosystem resilience while minimising damage to co-existing fire-sensitive species.
- Implement fire prevention strategies, including planned fuel treatments to decrease flammability by changing vegetation structure and/or changing vegetation composition. It is important to use ecologically sound strategies and minimise edge effects in fire-sensitive ecosystems[116].
- Co-develop projects for carbon sequestration through sustainable forest management and carbon trading mechanisms, but considering ecological criteria and social justice and principles of equitable sharing of benefits[117].

5. *Monitoring, evaluation, and adaptive management*:
- Establish monitoring and evaluation frameworks to assess the effectiveness and impacts of IFM actions incorporating feedback mechanisms to adapt strategies based on new insights, such as the unforeseen impacts of early season burning on gas and particulate matter emissions[118].

- Use adaptive management principles to adjust strategies and actions based on new information, changing conditions, and stakeholder feedback.

6. *Knowledge sharing and learning exchange:*
- Facilitate knowledge sharing and learning exchange among regions and countries to promote best practices in IFM implementation. It is important to ensure the traceability and due recognition of the local and regional knowledge by providing appropriate means and channels for knowledge transfer.
- Enhance local cooperation and knowledge sharing across government agencies, communities, Indigenous Peoples, non-governmental organisations, and other relevant stakeholders.
- Foster collaboration with international organisations, research institutions, and relevant networks to access technical expertise and support capacity building efforts ensuring that emerging challenges like unexpected air quality issues from prescribed burns are addressed.

7. *Review and re-assess:*
- Review and update the IFM strategy and action plan periodically based on evolving priorities, lessons learned, and emerging challenges.
- Improve IFM implementation continuously through interactive learning, adaptation, and innovation emphasising the importance of transparency about the potential negative impacts to foster a robust, adaptable management strategy.

## Incremental application of the proposed Roadmap

Transitioning from fire suppression systems to IFM systems at national level, as outlined in the proposed Roadmap (Fig. 2) requires significant policy reforms, and the adoption of new strategies at institutional, governance and social levels. Countries like Brazil and Mexico are successfully implementing IFM by shifting policy, delineating governance frameworks and promoting multi-stakeholder collaborations. However, because of its complexity, the implementation of the Roadmap has to be made through incremental steps to ensure feasibility and allow the flexibility required to adapt to local and regional contexts.

This incremental process takes existing local IFM practices as a baseline (Fig. 1, Supplementary Table S1). Local or regional centres of IFM practices have a fundamental role in setting the knowledge-science basis and producing scalable and realistic expertise for the implementation of the Roadmap. Local examples contribute to demonstrating the success of implementing IFM practices in local and regional socio-ecological systems and landscapes and generate opportunities for replication. They also provide a proof of concept for the effectiveness of IFM practices, and contribute to building a robust scientific understanding of IFM for implementation at a wider scale and in other contexts. Further development of IFM practices should involve establishing hubs for exchanging knowledge, learning together, building capacity and testing new context-specific approaches and involving members of the local population to raise awareness and trigger changes in societal perception about fire uses[81,90].

The path to escalate from local implementation of IFM practices to a widespread approach relies on implementation of steps 1–7 of the Roadmap (Fig. 2). To achieve this, countries must leverage their local and regional expertise and experience, and complement this with experiences and scientific insights from other regions and internationally. This can be supported through the enhancement of knowledge networks.

Although updating the legal frameworks is a fundamental step to formalise the IFM implementation, the multiple bottom-up actions discussed here contribute to advance the implementation of IFM under the current policy frameworks and without requiring specific IFM policies. However, effective governance structures and process facilitators are essential for coordinating actions and collaborations across multiple levels. The widespread implementation of IFM at regional and national scales depends critically on step 2 of the Roadmap 'Develop a policy and legal framework'[35]. The practitioner community implementing IFM practices on the ground must have that legal support to ensure the sustainability of the local IFM practices over time, to upscale local knowledge and to develop region or country-wide policies and strategies. Countries need to ensure that there are appropriate national and supra-national structures to ensure funding and infrastructure are available.

## Overcoming challenges for implementing IFM: future lines of research

The success of IFM relies on establishing a dialogue between traditional knowledge, land managers, fire science and fire policy through a strong collaboration between actors from different sectors and disciplines supported by boundary-spanning organisations. This dialogue is necessary to address simultaneously the ecological, economic, and social aspects of fires while recognizing the complexities of integrating actions across spatial and temporal scales and different governance systems (Table 1).

Legal and normative frameworks must recognize the socio-ecological role of fire and enable adaptive management policies. These frameworks should aid allocating economic resources toward building resilient landscapes through prevention, adaptation, and mitigation actions. Along with engaging local communities, adequate legal frameworks and economic resources can support adaptive vulnerability assessments and translate these into effective strategies for building capacity in local populations. Effectiveness studies are needed to ensure policies remain flexible and responsive to evolving conditions and knowledge bases.

Clear metrics and objectives are important for measuring the success and impact of IFM approaches. IFM effectiveness also relies heavily on ongoing research and adaptation to changing conditions, which requires significant efforts for which resources need to be allocated (Table 1). A forward-looking approach is essential, with priorities and future directions emphasising the development of dynamic, data-driven models for fire risk assessment. These models should leverage real-time environmental data and predictive analytics (scenarios), complemented by local capacity building to identify new fire-prone areas, including those where extreme fire behaviour is likely. Enhancing the ability of fire behaviour models to describe and predict extreme phenomena is also necessary. Planning for post-fire recovery, understanding the ecology of areas that are becoming fire-prone, and integrating research that combines fire likelihood with hydrology and erosion models require more ground data and investment in long-term research programs (Table 1).

While IFM often uses controlled, low-intensity burns to manage vegetation and reduce wildfire risk, the increasing risk of high-intensity burns - including both uncontrolled wildfires and "escaped" prescribed or planned burns - poses significant challenges. These intense fires can destroy soil seed banks, deplete essential nutrients, and alter soil structure, which inhibits natural regeneration and increases erosion risks. Additionally, they generate higher levels of greenhouse gas emissions and particulate matter, impacting air quality and contributing to climate change. Recent studies link the rising frequency and intensity of these fires to climate change, with cascading effects on biodiversity and ecosystem function[119]. To address when and where fire is an appropriate tool, IFM practices must prioritize carefully timed, low-intensity burns and incorporate adaptive management strategies that account for evolving climate conditions and the growing challenges of conducting safe, planned burns.

IFM approaches should also include justice aspects, especially when dealing with climate financing markets such as carbon abatements. Conflicting objectives, such as the negative impacts on biodiversity of early season burning plans needed to meet carbon abatement goals and reduce wildfire risk, need to be addressed through multi-stakeholder knowledge sharing and cooperation. Fostering multi-actor cooperation and collaboration, alongside legal frameworks that support adaptive management policies will be crucial. Effective stakeholder collaboration and knowledge sharing, community-based capacity building and empowerment, and the integration of IFM into broader climate change policies are imperative to overcome the current challenges to the implementation of IFM.

## Conclusions

IFM incorporates ecological and socio-cultural dimensions into the management needs for wildfire prevention, response and recovery. As such, it can be an effective tool to manage and restore fire-prone ecosystems because it simultaneously addresses societal challenges with the use of an ecological process that has multiple benefits for landscapes, society and nature.

IFM adopts interconnected objectives that directly address the impacts of climate change on fire regimes, fire behaviour, landscape resilience, biodiversity, and social systems. By reducing the risk of uncontrolled wildfires, IFM helps protect carbon stocks, conserve biodiversity, and safeguard community assets while building ecosystem resilience to future changes in fire regimes and climate. This dual role positions IFM as an invaluable strategy within climate policy frameworks, especially in regions experiencing increasing fire frequency and intensity.

The alignment of IFM with climate resilience and sustainable development goals highlights its potential to be integrated into policy frameworks at both national and international levels. As countries seek scalable, multi-benefit strategies to manage climate risks, incorporating IFM into adaptation and mitigation policies offers a proactive path to protect ecosystems and communities alike.

This review proposes a roadmap for effective IFM implementation, highlighted some of the current challenges and identified priorities and future research directions. This roadmap is designed to help nations address immediate wildfire risks and contribute to efforts to adapt to new fire regimes, but will also promote climate change adaptation and provide mitigation benefits. Future progress towards broader adoption of IFM programs requires the acknowledgement of the need to build resilient landscapes and a resilient society that allows us to live with altered fire regimes.

## Data availability

Data sharing not applicable to this article as no datasets were generated or analysed during the current study.

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

## Acknowledgements

Authors are grateful to the FIRE-ADAPT project (EU grant HORIZON-MSCA-2021-SE-01086416) for support and to the FIRE-ADAPT Consortium for general discussion of ideas during study hubs and exchange secondments. I.O.M. and M.S.M. were supported by NERC grant NE/W00058X/1. P.M.F. was supported by National Funds from FCT - Portuguese Foundation for Science and Technology, under project UIDB/04033/2020 (https://doi.org/10.54499/UIDB/04033/2020). L.B. was supported by the wildE Horizon Europe (GAP- 101081251) project. B.A.B. was funded by the LANDMARC project (European Union's Horizon 2020 grant agreement No 869367). AR.was supported by the EU-funded 'Firelogue' project (Grant agreement ID: 101036534) and a 'Ramón y Cajal' fellowship program of the Spanish Ministry of Science and Innovation (RYC2022-036822-I). S.dM. was supported by a Serra Húnter fellowship provided by the Government of Catalonia (Generalitat de Catalunya). G.J.H. was supported by the International Development Research Centre (IDRC), Ottawa, Canada. The views expressed herein do not necessarily represent those of IDRC or its Board of Governors.M.W.J. was funded by the Natural Environment Research Council (NERC; NE/V01417X/1). D.A.P. was supported by the Colombia Ministry of Science and the General System of Royalties project BPIN 2020000100456.

## Author contributions

I.O.M. & N.P.G. had the initial idea and secured funding (FIRE-ADAPT, UE/MCSA-SE/101086416). I.O.M., D.A.P. & N.P.G. developed manuscript objectives and outline. I.O.M. wrote the first draft of the manuscript with significant contributions from D.A.P. and N.P.G. G.L.S., P.F., R.P.G. & S.P.H. led sections of the manuscript and provided several rounds of critical feedback to the manuscript. Figures were developed by I.O.M., A.H., M.S.M., N.P.G., S.P.H. with feedback from G.L.S., R.P.G., R.C., P.F., P.P., S.dM. D.A., B.B., V.B., L.B., R.C., L.G.G., G.H., M.W.J., V.I., A.M., R.M.F., F.M., C.P., P.P., A.R., and M.S.O. provided critical contributions supported by literature that improved the manuscript (study cases, examples and critical feedback to content).

## Competing interests

The authors declare no competing interests.

## Additional information

[1]AMAP, University of Montpellier, CIRAD, IRD, CNRS, INRAE, Montpellier, France. [2]Environmental Change Institute, School of Geography and the Environment, University of Oxford, Oxford, UK. [3]Pau Costa Foundation, Taradell, Spain. [4]Department of Agriculture, Forest and Food Sciences, DISAFA, University of Torino, Largo Paolo Braccini 4, Grugliasco, Torino, Italy. [5]Department of Science, Technology and Society, University School for Advanced Studies IUSS Pavia, Palazzo del Broletto, Piazza della Vittoria 15, 27100 Pavia, Italy. [6]School of Environmental Sciences, University of East Anglia, Research Park, Norwich NR4 7TJ, England. [7]Centre for the Research and Technology of Agro-Environmental and Biological Sciences (CITAB), Inov4Agro, University of Trás-os-Montes and Alto Douro, 5000-801 Vila Real, Portugal. [8]Animal Biology Lab & BioLand. Departament de Ciències Ambientals, Universitat de Girona, 17071 Girona, Spain. [9]Department of Evolutionary Biology, Ecology and Environmental Sciences & IRBIO, Universitat de Barcelona, 08028 Barcelona, Spain. [10]Departamento de Estudios Ambientales, Universidad Simón Bolívar,

Apartado 89000, Valle de Sartenejas, Caracas 1080, Venezuela. [11]COBRA Collective (CIC), Egham, UK. [12]UMR Art-Dev-5281, Paul Valéry- University of Montpellier 3, Montpellier, France. [13]Fondazione Centro Euro-Mediterraneo sui Cambiamenti Climatici, IAFES Division, Via De Nicola, 9, 07100 Sassari, Italy. [14]National Research Council, Institute of BioEconomy (CNR-IBE), Traversa La Crucca 3, 07100 Sassari, Italy. [15]Forest Science and Technology Centre of Catalonia, Solsona 25280, Spain. [16]Consejo Superior de Investigaciones Científicas, Cerdanyola del Vallès 08193, Spain. [17]CREAF, 08193 Cerdanyola del Vallès, Spain. [18]Tyndall Centre for Climate Change Research and the School of Global Development, Norwich Research Park, University of East Anglia, Norwich, UK. [19]Department of Agricultural and Forest Sciences and Engineering, University of Lleida, 25198 Lleida, Spain. [20]Instituto Chico Mendes de Conservação da Biodiversidade, Brasília, Brazil. [21]African Climate & Development Initiative, University of Cape Town, Cape Town, South Africa. [22]Fundación Amigos de la Naturaleza, Santa Cruz de la Sierra, Bolívia. [23]Woodwell Climate Research Center, Falmouth, MA, USA. [24]Centro de Investigación y Transferencia Rafaela (CONICET - Universidad Nacional de Rafaela), Santa Fe, Argentina. [25]Centro Nacional de Prevenção e Combate aos Incêndios Florestais - Prevfogo Instituto Brasileiro do Meio Ambiente e dos Recursos Naturais Renováveis (IBAMA), Brasilia, Brazil. [26]UMR CEFE, University of Montpellier, CNRS, EPHE, IRD, Montpellier, France. [27]Misión Biolóxica de Galicia, Consejo Superior de Investigaciones Científicas (MGB-CSIC), 15705 Santiago de Compostela, Spain. [28]Geography and Environmental Science, University of Reading, Whiteknights, Reading RG6 6AH, UK. [29]ECOLMOD, Universidad Nacional de Colombia, Bogotá, Colombia. ✉e-mail: imma.oliverasmenor@ird.fr

