## [Transparent Peer Review file · Communications Earth & Environment]

Integrated fire management as an adaptation and mitigation strategy to altered fire regimes

Corresponding Author: Dr Imma Oliveras Menor

Version 0:

Decision Letter:

Dear Dr Oliveras Menor,

Your manuscript titled "Integrated fire management as a climate change adaptation and mitigation strategy in fire-prone ecosystems" has now been seen by 2 reviewers, whose comments are appended below. You will see that they find your work of potential interest. However, they have raised quite substantial concerns that must be addressed. In light of these comments, we cannot accept the manuscript for publication in its current form, but would be interested in considering a revised version that fully addresses these serious concerns.

We hope you will find the reviewers' comments useful as you decide how to proceed. Should additional work allow you to address these criticisms, we would be happy to look at a substantially revised manuscript. If you choose to take up this option, please either highlight all changes in the manuscript text file, or provide a list of the changes to the manuscript with your responses to the reviewers. We expect that your revision needs to meet the following editorial thresholds:

1. Provide a clear and specific description of IFM as a realistic and implementable climate change adaptation and mitigation strategy.
2. Expand description to larger geographical extent.
3. Consider changing manuscript format from perspective to review or provide more insights / suggestions for future research.

When resubmitting, please provide a point-by-point response to the reviewers' comments. Please submit your responses as a separate file, distinct from your cover letter where you can add responses to the Editors' comments that you do not want to be made available to the reviewers. Word files are preferred. We recommend that any figures, tables or graphs that are included in the response to reviewers are also included in the main article or Supplementary Information.

If the revision process takes significantly longer than three months, we will be happy to reconsider your paper at a later date, as long as nothing similar has been accepted for publication at Communications Earth & Environment or published elsewhere in the meantime.

Please use the following link to submit your revised manuscript, point-by-point response to the reviewers' comments with a list of your changes to the manuscript text (which should be in a separate document to any cover letter), a tracked-changes version of the manuscript (as a PDF file) and any completed checklist:

Link Redacted

Please do not hesitate to contact us if you have any questions or would like to discuss the required revisions further. Thank you for the opportunity to review your work.

Best regards,

Yongqiang Liu, PhD
Editorial Board Member
Communications Earth & Environment

Alice Drinkwater, PhD
Associate Editor
Communications Earth & Environment

EDITORIAL POLICIES AND FORMAT

If you decide to resubmit your paper, please ensure that your manuscript complies with our editorial policies and complete and upload the checklist below as a Related Manuscript file type with the revised article:

Editorial Policy Policy requirements
(Download the link to your computer as a PDF.)

- Behavioural and social science
- Ecological, evolutionary & environmental sciences
- Life sciences

<https://www.nature.com/documents/nr-reporting-summary.zip>

For your information, you can find some guidance regarding format requirements summarized on the following checklist: (<https://www.nature.com/documents/commsj-phys-style-formatting-checklist-article.pdf>) and formatting guide (<https://www.nature.com/documents/commsj-phys-style-formatting-guide-accept.pdf>).

REVIEWER COMMENTS:

Reviewer #1 (Remarks to the Author):

General comments

This manuscript is focused on integrated fire management (IFM) and aims to be global in geographical extent. The work also attempts to build a case for IFM being part of strategies for climate adaptation and mitigation, but this link is not always clear. Although the topic is within the scope of the journal, the manuscript falls short making a compelling case for IFM as a climate change adaptation and mitigation strategy. Furthermore, I found the material presented to be better suited to a review article (eg, Line 310-330) than a perspective. The manuscript, despite presenting lots of interesting information, is not a true perspective piece that is « intended to stimulate discussion ».

The geographical extent of the manuscript seems constrained to anthropogenic landscapes of Europe, as well as parts of Australia, for the most part. The most fire-prone part of the world, the African savanna, is barely covered. In addition, many of the presented information appears to be mostly relevant to fire regimes characterized by small, moderate-intensity fires. I question if some of the presented information is truly relevant to areas that are sparsely populated or that have fire regimes characterized by high-intensity fires. I can certainly appreciate the challenge (read, impossibility) of covering everything in a short review or perspective. Therefore, I recommend that the authors either limit the focus to certain types of areas or fire regimes or, alternatively, say something about IFM (or lack of IFM) in different fire-regimes types.

Reading the first part of the manuscript, it is not clear to me what constitutes IFM with respect to this paper. I understand that the concept is somewhat vaguely defined and that there is no official 'IFM' stamp or internationally regulated program, as there would be for sustainable forestry practices. However, the authors should spell out specifically what they considered to be an IFM project. It would also be useful to know whether the examples used are backed by an 'adequate legal framework' (Line 245) or they are simply inferred. Also, 'Governance structures' regarding IFM is mentioned a few times. It would be useful to understand what this entails. Finally, the link to climate change adaptation and strategy is fairly weak.

Given that this manuscript is intended as a perspective, I was expecting more bold, outside-the-box ideas related to this aspect.

With respect to style, I found some of the information to be somewhat redundant (eg, the different activities or strategies associated with IFM). Also, the transition between some ideas is abrupt at times. Many of the sentence (eg, Line 111-114, 135-140) pack in too much information (ie, more than one main idea), which makes them difficult to follow and which probably obfuscates the intended message. Some tightening and streamlining of the text would improve its ease of reading. The numerous subsections further made the manuscript feel more like a review paper than a perspective.

Specific comments

Line 53 : 'events' is not necessary (applies throughout); 'fire behavior events' is quite awkward.

Line 55: Consider replacing 'will' by 'may', as this is a single study and, in addition, these projections have a high level of uncertainty.

Line 68: The decrease in area burned seems contradictory to some of the information presented in the first paragraph. Does this decreasing trend still hold in 2024 (I suspect not). If there is indeed a decrease in area burned, it should probably be mentioned in the first paragraph.

Line 83: I recommend replacing 'erosion' with 'reduction', 'decrease', 'curtailment' or another word with a more direct meaning.

Line 87: Replace 'harm' by 'consequence' or another word with a more direct meaning.

Line 89-92. This is the stated goal. I would recommend splitting this sentence in two to better lay out the different elements of the paper. Specifically, the last idea of the sentence seems a bit buried, though I suspect it is fairly central to the paper.

Line 103: The fuel buildup statement needs to be backed by a reference.

Line 107-110: The last part of the sentence is confusing (especially 'leading to a conflation of diverse fire types').

Line 135: A replacement of 'fire management' by some other expression (eg, 'protection from wildfire') to not say that 'integrated fire management deliver (...) management', which is redundant.

Line 142-144: This seems like repeated information. In fact, this paragraph is wordy and could be condensed.

Line 150: I recommend including the last part of this paragraph in the next paragraph.

Line 163: It seems that anything related to IFM would depend more on specific jurisdictions within a country than the country itself?

Line 182: It seems a bit misleading to say that IFM allows farmers to burn. Have they now been burning pretty much forever?

Line 193: I do not want to downplay the importance of these initiatives, but 220 km² is a rather small area.

Line 233: 'standalone' is not a single word.

Line 267: It seems that this statement should be tempered. Such a measure is assuredly highly uncertain.

Reviewer #2 (Remarks to the Author):

This paper addresses the globally escalating fire risk by offering an approach, IFM, which it is argued would substantially reduce the risk while also addressing a range of social and environmental issues. As the authors acknowledge, the idea of IFM in various forms has been around for a long time.

The argument is well made and there is little doubt that IFM as conceptualised in this paper could and occasionally does deliver the wide range of benefits discussed in the paper. However, large scale success with IFM would appear to require major changes to fundamental institutional, organisational, and in some cases social attitudes, and implementation would be more than challenging. The paper conclusion left me wondering whether the argument is a normative statement about IFM, or do the authors see their prescription as something that could be achieved in the world as it exists today? There was an issue in other parts of the paper as well. Could the authors clarify the extent to which the version of IFM they set out is achievable or is it an ideal? There are other related comments below.

Some detailed points follow. These are mostly on planned fire/prescribed burns, which is often key to wildfire risk reduction and landscape and ecosystem management, but which can also result in severe damage.

I would like the authors to consider these points:

Line 172 this comment on colonial legacies would likely also apply to many countries outside Africa. Can example be provided of countries that were not colonised and which now have different systems?

L 273 – what about very hot burns which can also destroy soil seed banks? These are becoming more common. A recently published analysis concerns wildfires, but escaped planned burns are subject to the same forces. (See this paper on fire intensity: Cunningham, C.X., Williamson, G.J. & Bowman, D.M.J.S. Increasing frequency and intensity of the most extreme wildfires on Earth. *Nat Ecol Evol* 8, 1420–1425 (2024). <https://doi.org/10.1038/s41559-024-02452-2>).

L266 – the comment that indigenous people manage 85% of the world's biodiversity - 80% has been the more widely quoted statistic. Please see a September 2024 paper in *Nature*: <https://www.nature.com/articles/d41586-024-02811-w>. This paper doesn't undermine the authors' argument for the value of traditional and indigenous knowledge, but shows that the 85% (more usually 80%) statistic is likely to be incorrect. Please amend the text or explain why the 85% figure is

retained.

L272-273 – "...enriches the soil" – this depends on the soil and the wildfire. "Cool" fires can do this, but higher intensity fires which are becoming much more common (see comment above), destroy vegetation, change soil attributes and structure and can result in massive erosion.

L282 – a point is made on losses avoided by protection of other ecosystem services through planned fire. However, there are also occasions when all types of fires result in severe ecosystem service disruption to air and water.

Line 260, section 2 - Planned fire/prescribed fire can do all these things. But often do not and in a hot drying world such fires are more difficult to manage, and can easily and sometimes do lead to severe smoke pollution and damaged reputations for tourist destinations; for example, see Johnston F.H. (2020) MJA 213 (6) 21 September.

Occasionally, there is also significant loss of all types including environmental loss. David Lindenmayer and colleagues have recently written about the limitations of planned burns in some landscape types and some types of fire regimes.

Section 3 – IFM can and sometimes does the things listed, although in practice there are many issues with implementation. In a hot dry world as mentioned above (or during wet periods) control of planned burns and ecological fires is challenging with a result that burns either cannot be undertaken or are likely to be too hot and damaging, as in the comment above. The paper acknowledges that the window for planned burns is narrowing, but this qualifier seems lost in the discussion of IFM.

Line 301 – land abandonment in Europe has led to increased fuel and decreased local capacity for land and fire management – this reflects fundamental social and economic changes and is probably unlikely to be reversed by IFM.

The Roadmap section – this section sets out a series of major requirements for IFM. If all are needed, it almost seems that IFM might not be achievable? Can the authors suggest prioritises or essential initial steps?

The conclusion is more of what could be rather than what is likely. Could something be said on how the necessary transition (to get to IFM) can be started?

Communications Earth & Environment is committed to improving transparency in authorship. As part of our efforts in this direction, we are now requesting that all authors identified as 'corresponding author' create and link their Open Researcher and Contributor Identifier (ORCID) with their account on the Manuscript Tracking System prior to acceptance. ORCID helps the scientific community achieve unambiguous attribution of all scholarly contributions. You can create and link your ORCID from the home page of the Manuscript Tracking System by clicking on 'Modify my Springer Nature account' and following the instructions in the link below. Please also inform all co-authors that they can add their ORCIDs to their accounts and that they must do so prior to acceptance.

Version 1:

Decision Letter:

Dear Dr Oliveras Menor,

Your manuscript titled "Integrated fire management as an adaptation and mitigation strategy to altered fire regimes" has now been seen by our reviewers, whose comments appear below. In light of their advice we are delighted to say that we are happy, in principle, to publish a suitably revised version in Communications Earth & Environment.

We therefore invite you to revise your paper one last time to address the remaining concerns of our reviewers. At the same time we ask that you edit your manuscript to comply with our format requirements and to maximise the accessibility and

therefore the impact of your work.

EDITORIAL REQUESTS:

****Please take care to match our formatting and policy requirements. We will check revised manuscript and return manuscripts that do not comply. Such requests will lead to delays. ****

SUBMISSION INFORMATION:

OPEN ACCESS:

Communications Earth & Environment is a fully open access journal. Articles are made freely accessible on publication. For further information about article processing charges, open access funding, and advice and support from Nature Research, please visit <https://www.nature.com/commsenv/open-access>

Link Redacted

Best regards,

Alice Drinkwater, PhD
Associate Editor
Communications Earth & Environment
@CommsEarth

REVIEWERS' COMMENTS:

Reviewer #1 (Remarks to the Author):

I am satisfied with how the authors have addressed my concerns with the first version of the manuscript. Making this a review article rather than a perspective is appropriate and has helped the authors strength many aspects of the story. The examples are also more compelling, on the whole, than in the previous version. I have included below some minor suggestions for their consideration.

Line 65--Use either 'significant' or 'large', not both

Line 84-88--This is a paragraph composed of a single sentence that is difficult to follow. Could it be broken down and perhaps incorporated to the last paragraph? More importantly, it jumps rather abruptly into the concept of IFM without providing a definition (which could be brief, knowing that it will be further developed in following section).

Line 95--There is still somewhat of a western bias in this section. Many of these issues do not really apply some highly fire-prone parts of the world. However, as I had mentioned previously, I can appreciate that not every aspect of global wildfires can be covered in a single review article. A quick fix may simply to highlight where the described situations or phenomena are occurring in the world.

Line 99--United States could be replaced by 'North America'. Many of the technical innovations for fire fighting were

developed and patented in Canada, including some of the more well-used water bombers, which are mentioned in the following sentence.

Line 144-146--But are those the main reasons that '... kept IFM initiatives at relatively small scales'? The lack of political will would top my list.

Line 246-249--This short paragraph seems a bit detached from the previous one. I suggest merging it to some other paragraph to improve the flow.

Line 356-362--This paragraph covers an important topic that remains hotly debated, given the unequal effectiveness shown by fuel treatments during some wildfire catastrophes. I think we all agree that fuel treatments are essential, but they can have some harmful ecological consequences (eg, habitat fragmentation). This is not an obligation, but it could be interesting to mention that fuel treatments could be designed and maintained in a way this is not detrimental to other values and objectives, such as the maintenance of biodiversity or the protection/conservation of species of interest.

Line 513--Some of the material in this section seems to be repeating information provided in previous sections. The section couldn't be tightened to focus on a few, key future lines of research.

Reviewer #2 (Remarks to the Author):

The authors have responded constructively to the comments of both reviewers and I believe that the manuscript is now ready for publication. It is a very interesting and timely contribution, and I expect it will influence thinking in the field.

I have a few very minor observations for the authors to consider.

Line 74 – there is a typo: it should read “significantly”

Ln 314 – the use of aerials started well before the 1970s, for example, California started using them in the mid 1950s. That said, they were not a significant factor in fire-fighting until later.

Ln 526 – The example of Ethiopia is in a paragraph which starts with a comment on the colonial legacy, but earlier in the responses to comments, the authors state (correctly) that Ethiopia is an example of a country that was not colonised.

Ln 678 – the usual expression is “...the grass is greener...”

REVIEWER COMMENTS:

Reviewer #1 (Remarks to the Author):

General comments

This manuscript is focused on integrated fire management (IFM) and aims to be global in geographical extent. The work also attempts to build a case for IFM being part of strategies for climate adaptation and mitigation, but this link is not always clear. Although the topic is within the scope of the journal, the manuscript falls short making a compelling case for IFM as a climate change adaptation and mitigation strategy. Furthermore, I found the material presented to be better suited to a review article (eg, Line 310-330) than a perspective. The manuscript, despite presenting lots of interesting information, is not a true perspective piece that is « intended to stimulate discussion ».

We appreciate the reviewer's observation. One of the main challenges– and the reason we initially submitted it as a perspective – is that we found very limited examples where IFM is applied as a CC strategy, and these few examples remain local and context-dependent. We aimed thus at saying that this is a missed opportunity and that, besides many other benefits (ecological, societal), IFM can be a NbS for climate change adaptation and mitigation.

After careful consideration and discussion with co-authors, and following the reviewer's recommendation, we now converted the manuscript to a review in which we synthesize the advances and challenges for IFM, and we set goals and a roadmap to aid stake-holders to implement IFM programs with multiple benefits. Therefore, we have changed the title to: 'Integrated fire management as an adaptation and mitigation strategy to altered fire regimes'.

We hope these changes provide a more balanced piece on what IFM is, what are its benefits to People and Nature, the state-of-art in implementation, and our proposed framework to be adopted as a program in places impacted by altered fire regimes.

Nonetheless, this manuscript has quite a few elements that advance thinking and foster discussion (e.g. the roadmap and proposed implementation), so if the Editor disagrees with this change, we would be happy to re-consider.

The geographical extent of the manuscript seems constrained to anthropogenic landscapes of Europe, as well as parts of Australia, for the most part. The most fire-prone part of the world, the African savanna, is barely covered. In addition, many of the presented information appears to be mostly relevant to fire regimes characterized by small, moderate-intensity fires. I question if some of the presented information is truly relevant to areas that are sparsely populated or that have fire regimes characterized by high-intensity fires. I can certainly appreciate the challenge (read, impossibility) of covering everything in a short review or perspective. Therefore, I recommend that the authors either limit the focus to certain types of areas or fire regimes or, alternatively, say something about IFM (or lack of IFM) in different fire-regimes types.

We appreciate this feedback. Although we tried to keep a broad perspective (Figure 1 and Table S1), literature is inherently biased. We have added more examples of IFM practices from

underrepresented regions, such as Zambia (line 1740177), Ethiopia (lines 177-182), and Southeast Asia (lines 206-217). We have also updated the examples from Latin America (lines 183-205) and incorporated examples from all regions throughout the text.

We also expanded Figure 1 and Table S1 to include more diverse global examples of IFM practices across varied fire regimes. These additions reinforce the manuscript's global perspective and demonstrate the wide applicability of IFM in different ecological and socio-economic contexts.

Reading the first part of the manuscript, it is not clear to me what constitutes IFM with respect to this paper. I understand that the concept is somewhat vaguely defined and that there is no official 'IFM' stamp or internationally regulated program, as there would be for sustainable forestry practices. However, the authors should spell out specifically what they considered to be an IFM project. It would also be useful to know whether the examples used are backed by an 'adequate legal framework' (Line 245) or they are simply inferred. Also, 'Governance structures' regarding IFM is mentioned a few times. It would be useful to understand what this entails. Finally, the link to climate change adaptation and strategy is fairly weak. Given that this manuscript is intended as a perspective, I was expecting more bold, outside-the-box ideas related to this aspect.

We appreciate this important point and have revised the manuscript to clarify that while IFM is an ambitious framework, incremental steps can make its implementation feasible. We have updated and incorporated recent advances in IFM (like the Brazil Law on Integrated Fire Management – line 189, and FAO's updated IFM Voluntary Guidelines -line 273).

We have revised the manuscript to provide a more precise definition of IFM (following Myers et al 2006), clearly outlining its core objectives, and how the proposed roadmap can lead to effective IFM programmes. Additionally, we clarified how legal frameworks and governance structures underpin IFM initiatives, using concrete examples of successful implementations supported by local and national policies (e.g. section 'Advances in developing and implementing IFM; point (2) of Roadmap).

With respect to style, I found some of the information to be somewhat redundant (eg, the different activities or strategies associated with IFM). Also, the transition between some ideas is abrupt at times. Many of the sentence (eg, Line 111-114, 135-140) pack in too much information (ie, more than one main idea), which makes them difficult to follow and which probably obfuscates the intended message. Some tightening and streamlining of the text would improve its ease of reading. The numerous subsections further made the manuscript feel more like a review paper than a perspective.

We have made a thorough revision of the manuscript, re-ordered (and reduced) subsections and deleted parts that were redundant. Our English-native co-authors have done tightening and streamlining of the text to improve clarity.

Specific comments

Line 53 : 'events' is not necessary (applies throughout); 'fire behavior events' is quite awkward.

Done

Line 55: Consider replacing 'will' by 'may', as this is a single study and, in addition, these projections have a high level of uncertainty.

Done

Line 68: The decrease in area burned seems contradictory to some of the information presented in the first paragraph. Does this decreasing trend still hold in 2024 (I suspect not). If there is indeed a decrease in area burned, it should probably be mentioned in the first paragraph.

We have deleted that in the text as it was not relevant to the paper. A recent paper that covers period 2001-2023 led by one of the co-authors (M. Jones et al Science – Fig. 3 pasted here for an easier discussion) finds a slight global increase in forest burned area but a strikingly 167% increase in burned area in the regions that comprises Eurasian boreal regions, western North America (e.g., Sierra Nevada forests, North-Central Rockies forests, Muskwa-Slave lake forests, Fraser Plateau and Basin complex, and Northwest Territories taiga), Chile (Valdivian temperate forests), and China (Northeast China Plain deciduous forests and Hengduan Mountains conifer forests).

To reflect their latest findings, including a 60% increase in forest fire carbon emissions, we have slightly rephrased that paragraph in the introduction (lines 64-73).

Jones, M. W. et al. (2024). Global rise in forest fire emissions linked to climate change in the extratropics. *Science*, 386(6719), ead15889

Line 83: I recommend replacing 'erosion' with 'reduction', 'decrease', 'curtailment' or another work with a more direct meaning.

We have changed 'erosion' to 'disappearance'.

Line 87: Replace 'harm' by 'consequence' or another work with a more direct meaning.

Done

Line 89-92. This is the stated goal. I would recommend splitting this sentence in two to better lay out the different elements of the paper. Specifically, the last idea of the sentence seems a bit buried, though I suspect it is fairly central to the paper.

Thank you for this useful suggestion. We have now rephrased our stated goal to: *'This review examines current fire management practices, with a focus on Integrated Fire Management (IFM) as an adaptation and mitigation strategy to altered fire regimes. We review the concept of IFM, assess the progress and challenges in its implementation across different regions worldwide. We then propose five core objectives and a roadmap of incremental steps for implementing IFM as a strategy to adapt to ongoing and future changes in fire regimes, and maximise the potential of IFM to provide benefits to people and nature'* (lines 88-94).

Line 103: The fuel buildup statement needs to be backed by a reference.

We have added the following reference:

Carroll MS, Blatner KA, Cohn PJ, Keegan CE, Morgan T. 2007. Managing fire danger in the forests of the US inland northwest: A classic "wicked problem" in public land policy. *Journal of Forestry* 105: 239–244.

Line 107-110: The last part of the sentence is confusing (especially 'leading to a conflation of diverse fire types').

We have reworded the sentence to: *'Socio-economic and land-use changes in the 20th century—such as rural abandonment (i.e., migration from rural to urban areas), the growth of industrial forestry and agribusiness, and the expansion of residential areas increasing wildland-urban interfaces—have strengthened the perception of fire as a universal threat to society and ecosystems. This has led to a widespread consensus that fire should be suppressed at all costs, regardless of the type of fire'* (lines 103 – 108).

Line 135: A replacement of 'fire management' by some other expression (eg, 'protection from wildfire') to not say that 'integrated fire management deliver (...) management', which is redundant.

We have changed to *'The concept of IFM defined by Myers et al. (2006) has gradually gained recognition among fire practitioners and scientists because of its potential to deliver effective, efficient and equitable fire management'* (lines 134-136).

Line 142-144: This seems like repeated information. In fact, this paragraph is wordy and could be condensed.

We agree with the reviewer that this was redundant, and have deleted the sentence.

Line 150: I recommend including the last part of this paragraph in the next paragraph.

Done

Line 163: It seems that anything related to IFM would depend more on specific jurisdictions within a country than the country itself?

We acknowledge that the full-scale implementation of IFM as presented would indeed require substantial changes at institutional, organizational, and social levels, particularly in terms of governance structures, resource allocation, and community engagement. However, we do not intend for IFM to be seen solely as an idealistic or aspirational framework. Rather, we view IFM as a realistic and implementable strategy, but one that must be introduced incrementally and that certainly needs to acknowledge the local context, and avoid governance structured aimed at 'one-size fits all' .

We have reworded several parts of the manuscript to reflect this, and added a sub-section on how the roadmap can be implemented (lines 474 – 510). Initial steps, such as engaging communities, building capacity, and initiating policy reform, are now highlighted as foundational actions that can support gradual scaling of IFM. This revised roadmap demonstrates that IFM can be implemented progressively, allowing regions to start with manageable actions while building toward more comprehensive programs over time.

Line 182: It seems a bit misleading to say that IFM allows farmers to burn. Have they now been burning pretty much forever?

This sentence refers to the specific example Kafue National Park in Zambia. Farmers were before not allowed to legally use fire, while they are now. We have rephrased the sentence to leave that clearer (lines 173-177).

Line 193: I do not want to downplay the importance of these initiatives, but 220 km² is a rather small area.

We are uncertain of what the reviewer's point here is, so we are not acting on this comment.

Line 233: 'standalone' is not a single word.

This word is now deleted.

Line 267: It seems that this statement should be tempered. Such a measure is assuredly highly uncertain.

Indeed, while our manuscript was in review a new analysis on Indigenous People's lands was published and provided updated numbers and facts. We have rephrased this sentence to accordingly reflect the latest scientific evidence: '*It is estimated that Indigenous Peoples steward or hold tenure rights over at least of 37% of the Earth's terrestrial natural lands, which contain more than one-third of the world's intact forests (Fernández-Llamazares et al 2024)*'. (Lines 303-305).

Fernández-Llamazares et al (2024). A baseless statistic could harm the Indigenous Peoples it is meant to support. *Nature*, vol 633, 32-35. Published 5 September 2024, corrected 10 October 2024.

Reviewer #2 (Remarks to the Author):

This paper addresses the globally escalating fire risk by offering an approach, IFM, which it is argued would substantially reduce the risk while also addressing a range of social and environmental issues. As the authors acknowledge, the idea of IFM in various forms has been around for a long time.

The argument is well made and there is little doubt that IFM as conceptualised in this paper could and occasionally does deliver the wide range of benefits discussed in the paper. However, large scale success with IFM would appear to require major changes to fundamental institutional, organisational, and in some cases social attitudes, and implementation would be more than challenging. The paper conclusion left me wondering whether the argument is a normative statement about IFM, or do the authors see their prescription as something that could be achieved in the world as it exists today? There was an issue in other parts of the paper as well. Could the authors clarify the extent to which the version of IFM they set out is achievable or is it an ideal? There are other related comments below.

The reviewer raises an important question about whether our conceptualization of IFM is a normative statement or an achievable strategy under current conditions. We acknowledge that the full-scale implementation of IFM as presented would indeed require substantial changes at institutional, organizational, and social levels, particularly in terms of governance structures, resource allocation, and community engagement. However, we do not intend for IFM to be seen solely as an idealistic or aspirational framework. Rather, we view IFM as a realistic and implementable strategy, but one that must be introduced incrementally. While large-scale adoption poses challenges, there are already numerous examples of smaller-scale success stories in fire-prone regions and, to the best of our knowledge, a few national IFM policy frameworks with the adequate legal regulation (e.g. Mexico and Brazil). These cases demonstrate that when appropriately supported by legal frameworks, financial resources, and stakeholder collaboration, many elements of IFM can be implemented effectively. To emphasize this, we have expanded the examples of Mexico and Brazil (lines 189 – 197), as well as added a subsection on how the roadmap can be implemented in incremental steps (lines 474 – 510).

Some detailed points follow. These are mostly on planned fire/prescribed burns, which is often key to wildfire risk reduction and landscape and ecosystem management, but which can also result in severe damage.

I would like the authors to consider these points:

Line 172 this comment on colonial legacies would likely also apply to many countries outside Africa. Can example be provided of countries that were not colonised and which now have different systems?

Thank you for this remark. We have performed a search among the very few countries that were not European-colonized nor were a colonizing power at some point in history. Ethiopia came across the only relevant example in which there are efforts to develop a national IFM strategy, although progress has not left the paper and fire policies are still fully focused on fire suppression and on prohibiting fire uses. (Lines 177-182).

H
Y vol. 61 169–174.

P

E

R273 – what about very hot burns which can also destroy soil seed banks? These are becoming more common. A recently published analysis concerns wildfires, but escaped planned burns are subject to the same forces. (See this paper on fire intensity: Cunningham, C.X., Williamson, G.J. & Bowman, D.M.J.S. Increasing frequency and intensity of the most extreme wildfires on Earth. *Nat Ecol Evol* 8, 1420–1425 (2024). <https://doi.org/10.1038/s41559-024-02452-2>).

"

We agree that the risk of high-intensity burns is significant, particularly under climate change. The manuscript has been updated to acknowledge that extreme burns, whether due to uncontrolled wildfires or escaped planned burns, pose significant risks to soil health, particularly by destroying soil seed banks and altering nutrient composition. We have added a paragraph discussing these risks, referencing recent findings (Cunningham et al., 2024) on the growing frequency of high-intensity fires (lines 537 – 544). This addition emphasizes the importance of adaptive management within IFM to prevent unintended high-intensity burns and mitigate their impacts on biodiversity and soil resilience.

a

p

L266 – the comment that indigenous people manage 85% of the world's biodiversity - 80% has been the more widely quoted statistic. Please see a September 2024 paper in Nature: <https://www.nature.com/articles/d41586-024-02811-w>

This paper doesn't undermine the authors' argument for the value of traditional and indigenous knowledge, but shows that the 85% (more usually 80%) statistic is likely to be incorrect. Please amend the text or explain why the 85% figure is retained.

d

a

g

r

o

f

o

Thank you for this relevant remark. This study came out while our manuscript was in review, and we have now changed this sentence to reflect the findings on this publication to: *'It is estimated that Indigenous Peoples steward or hold tenure rights over at least of 37% of the Earth's terrestrial natural lands, which contain more than one-third of the world's intact forests (Fernández-Llamazares et al 2024)'*. (Lines 303-305).

Fernández-Llamazares et al (2024). A baseless statistic could harm the Indigenous Peoples it is meant to support. *Nature*, vol 633, 32-35. Published 5 September 2024, corrected 10 October 2024.

L272-273 – "...enriches the soil" – this depends on the soil and the wildfire. "Cool" fires can do this, but higher intensity fires which are becoming much more common (see comment above), destroy vegetation, change soil attributes and structure and can result in massive erosion.

Thank you. As we have mentioned above, this is an important point that we have now rephrased to reflect better in the text. We are proposing IFM core objectives, and within those would be the need to promote 'cool' fires (i.e. low intensity) through IFM practices in order to avoid 'hot' fires through uncontrolled wildfires. (lines 306 -315).

L282 – a point is made on losses avoided by protection of other ecosystem services through planned fire. However, there are also occasions when all types of fires result in severe ecosystem service disruption to air and water.

We have added text (lines 244-247) discussing the potential negative impacts of all fire types, including their effects on air and water quality, thus presenting a balanced view of the consequences of fire use. We have also added references on using planned burns outside the ecological burning season (lines 248-260)

Line 260, section 2 - Planned fire/prescribed fire can do all these things. But often do not and in a hot drying world such fires are more difficult to manage, and can easily and sometimes do lead to severe smoke pollution and damaged reputations for tourist destinations; for example, see Johnston F.H. (2020) *MJA* 213 (6) 21 September.

Occasionally, there is also significant loss of all types including environmental loss. David Lindenmayer and colleagues have recently written about the limitations of planned burns in some landscape types and some types of fire regimes.

We have expanded the discussion on the difficulties of conducting prescribed burns in the 'The grass is not always green' section. We have also provided additional context on how planned burns may exacerbate certain risks in a warmer and drier world (Table 1, lines 576- 583).

Finally, we have also emphasized in the text that IFM I not only about prescribed or planned burning, sometimes fire is not to be used (lines 424, line 584)

Section 3 – IFM can and sometimes does the things listed, although in practice there are many issues with implementation. In a hot dry world as mentioned above (or during wet periods)

control of planned burns and ecological fires is challenging with a result that burns either cannot be undertaken or are likely to be too hot and damaging, as in the comment above. The paper acknowledges that the window for planned burn is narrowing, but this qualifier seems lost in the discussion of IFM.

Thank you for this very good point. We have added a section 'The grass is not always green' in which we discuss some of the limitations regarding fire management and progress on adopting more holistic approaches of IFM (as we intend in this paper) that do not necessarily involve the use of fire. This is also clarified in other parts of the text

Line 301 – land abandonment in Europe has led to increased fuel and decreased local capacity for land and fire management – this reflects fundamental social and economic changes and is probably unlikely to be reversed by IFM.

We have added a paragraph about the challenges on implementing IFM in Europe – not only what the reviewer raises, also we lack knowledge on what traditional fire uses were and to what extent they've survived (Lines 264-271).

The Roadmap section – this section sets out a series of major requirements for IFM. If all are needed, it almost seems that IFM might not be achievable? Can the authors suggest prioritises or essential initial steps?

We have added a section of incremental steps to implement the Roadmap (lines 474-510)

The conclusion is more of what could be rather than what is likely. Could something be said on how the necessary transition (to get to IFM) can be started?

We have reworded the conclusions